# Gadolinium Neutron Capture Therapy for Cats and Dogs with Spontaneous Tumors Using Gd-DTPA

**DOI:** 10.3390/vetsci10040274

**Published:** 2023-04-04

**Authors:** Vladimir Kanygin, Alexander Zaboronok, Aleksandr Kichigin, Elena Petrova, Tatyana Guselnikova, Andrey Kozlov, Dmitriy Lukichev, Bryan J. Mathis, Sergey Taskaev

**Affiliations:** 1Budker Institute of Nuclear Physics, Siberian Branch of Russian Academy of Sciences, ave. Lavrentiev, 11, 630090 Novosibirsk, Russia; 2Laboratory of Nuclear and Innovative Medicine, Department of Physics, Novosibirsk State University, Pirogov str., 1, 630090 Novosibirsk, Russia; 3Department of Neurosurgery, Faculty of Medicine, University of Tsukuba, Tennodai 1-1-1, Tsukuba 305-8575, Ibaraki, Japan; 4Veterinary Clinic “Best”, Frunze str., 57, 630005 Novosibirsk, Russia; 5Nikolaev Institute of Inorganic Chemistry, Siberian Branch of Russian Academy of Sciences, ave. Lavrentiev, 3, 630090 Novosibirsk, Russia; 6Clinical Hospital “Avicenna”, Uritskogo str., 2, 630007 Novosibirsk, Russia; 7International Medical Center, University of Tsukuba Hospital, Amakubo 2-1-1, Tsukuba 305-8576, Ibaraki, Japan

**Keywords:** veterinary medicine, cancer, spontaneous tumors, large animal irradiation, gadolinium neutron capture therapy, GdNCT, dimeglumine gadopentetate, Gd-DTPA, boron neutron capture therapy, BNCT

## Abstract

**Simple Summary:**

Neutron capture therapy, or NCT, is a unique irradiation-based method that is used to treat invasive advanced tumors as it minimizes the impact on healthy cells and tissues. Modern neutron sources for NCT include accelerators that can be installed in treatment facilities. One of the main conditions for the successful application of NCT is sufficient accumulation of the drug in tumor cells which allows it to capture neutrons, be activated, and release energy that destroys cancer cells. Among the elements capable of capturing neutrons, boron, gadolinium, and lithium are considered to be the most suitable for NCT. In our study, we showed the results of neutron capture therapy with gadolinium in a group of dogs and cats with spontaneous tumors to clarify the technical aspects of the method while adjusting the treatment conditions as close as possible to those of clinical trials.

**Abstract:**

We conducted a clinical veterinary study on neutron capture therapy (NCT) at a neutron-producing accelerator with seven incurable pets with spontaneous tumors and gadolinium as a neutron capture agent (gadolinium neutron capture therapy, or GdNCT). Gadolinium-containing dimeglumine gadopentetate, or Gd-DTPA (Magnevist^®^, 0.6 mL/kg b.w.), was used. We observed mild and reversible toxicity related to the treatment. However, no significant tumor regression in response to the treatment was observed. In most cases, there was continued tumor growth. Overall clinical improvement after treatment was only temporary. The use of Gd-DTPA for NCT had no significant effects on the life expectancy and quality of life of animals with spontaneous tumors. Further experiments using more advanced gadolinium compounds are needed to improve the effect of GdNCT so that it can become an alternative to boron neutron capture therapy. Such studies are also necessary for further NCT implementation in clinical practice as well as in veterinary medicine.

## 1. Introduction

Certain malignancies are extremely difficult to treat and can still remain lethal following treatment due to both their invasive nature and the inability of all modern treatment modalities to influence single tumor cells that migrate into healthy tissues and lead to the development of new tumor masses at a distance from the initial tumor that seemed to be cured. This may happen in the case of malignant melanoma with distant metastases, malignant glioma with fast local recurrence, or another type of invasive cancer that leaves no chance of survival despite the best efforts of modern medicine. Therefore, the search for more effective cancer therapies, including more advanced irradiation methods capable of targeting single tumor cells, is still ongoing.

The principle of low-energy neutron capture by a drug followed by a local release of energy that destroys tumor cells has been employed in neutron capture therapy (NCT), which is a unique and sometimes the only applicable treatment for certain types of malignancies. The main advantage of NCT is its selectivity such that only tumor cells are affected and healthy tissues are preserved due to the precise intracellular accumulation of a compound that captures neutrons, decays, and produces energy that disrupts tumor cell DNA integrity [1,2,3]. Several elements have been considered as neutron capture agents. boron-10 (^10^B)-based compounds have already been tested clinically, with one of them, boronophenyl alanine (BPA), finally approved for clinical application in Japan [4,5]. The gadolinium-152 isotope (^157^Gd) possesses the best ability to capture neutrons and releases gamma rays that ionize matter in tumor cells [6]. Experiments on lithium-6 (^6^Li) used as a neutron capture compound for NCT have been recently initiated, and additional details related to its application are yet to be discovered [7].

In the case of ^10^B, a concentration greater than 20 µg/g in tumor tissue is needed to achieve a therapeutic effect [1,2]. The optimal Gd load for successful GdNCT was estimated to be 50–200 µg/g in wet tumor tissue [6]. According to the recommendations, these two elements’ tumor/normal tissue concentration ratios should be ~3 or higher during irradiation [1]. ^10^B has a smaller neutron capture cross-section and the energy released due to its interaction with a neutron is limited by cell size, which plays a key role in treatment selectivity. Clinical trials on NCT at nuclear reactors using ^10^B (boron neutron capture therapy, or BNCT) have demonstrated effectiveness in treating different malignancies [8,9,10,11,12,13,14]. For example, Fukuda et al. showed the effectiveness and limitations of BNCT for malignant melanoma [8]; Kato et al. [9] and Kankaanranta et al. [12] reported on initial clinical trials on BNCT for recurrent head and neck cancer; Japanese research groups, including Yamamoto et al. [11] and Miyatake et al. [13], demonstrated the efficacy of the method in patients with malignant brain tumors; and Wittig et al. showed the ability of ^10^B compounds to accumulate in liver metastases of colorectal cancer [10].

However, the efficacy limitations of previously and currently used boron compounds, the complexity of the boron enrichment process required for their production, and their high cost restrict the use of ^10^B-containing compounds in research and clinical practice. Therefore, Gd-based drugs, which are already registered in many countries and used clinically for purposes other than NCT, could be alternatives to ^10^B compounds.

Thus, beginning in 1985, dimeglumine gadopentetate (Gd-DTPA), or Magnevist^®^, became one of the first contrast agents to be used in clinical trials and was originally approved by the FDA for clinical use in 1988 [15,16,17]. In magnetic resonance imaging (MRI), contrast agents are used for tumor tissue visualization improvement due to their accumulation in tumors, leading to an increased image signal. The increase in tumor tissue signal persists for about 1–2 h depending on the type of tumor, its blood supply, the contrast agent used, and its dosage. The pharmacokinetics, toxicity, and biodistribution of different gadolinium-based compounds used for contrast enhancement in MRI, including Gd-DTPA, have been extensively studied [18,19,20]. It is estimated that gadolinium concentrations as high as 300 µg ^157^Gd/g can be achieved in brain tumors with an MRI contrast agent such as Gd-DTPA when injected intravenously at 0.5 mmol/kg body weight [21]. The effectiveness of Gd compounds in NCT has been predicted in numerous studies examining their accumulation and biodistribution using tumor cell lines and in tumor-bearing laboratory animals [22,23,24,25,26,27,28,29]. 

Regarding the irradiation facility for NCT, a specialized nuclear reactor channel or a charged particle accelerator is needed for neutron generation [30,31,32,33]. For various reasons, nuclear reactor-based NCT has ceased to exist in many countries except for clinical and preclinical trials in Taiwan, Argentina, and China and new compound development and other research purposes in Japan [14,34,35,36,37,38]. Thus, improving NCT relies on the results of state-of-the-art accelerator research being conducted or planned in Japan, China, South Korea, Finland, Argentina, and Russia [32,33,39,40,41,42,43]. Among the most advanced projects, we can mention the BNCT System NeuCure^®^ by Sumitomo Heavy Industries, Ltd. (Tokyo, Japan), based on a cyclotron accelerator operated at an energy level of 30 MeV with a current of 2 mA and a beryllium target, which is already installed in clinical treatment facilities in Japan [3,5,32]. A tandem accelerator with vacuum insulation and a solid lithium target has been developed at the Budker Institute of Nuclear Physics in Novosibirsk, Russian Federation and later, together with TAE Life Sciences (Foothill Ranch, CA, USA), modified and installed in Xiamen Humanity Hospital (Xiamen, Fujian, China) [3]. The University of Tsukuba, together with Mitsubishi Heavy Industry Co., Ltd. (Tokyo, Japan) and several other organizations, have developed a linear accelerator with a beryllium target that has been tested preclinically, with plans to initiate clinical trials in 2023 [3,33]. A BNCT center in Helsinki, Finland, with an Hyperion™ electrostatic accelerator with a rotating lithium target developed by Neutron Therapeutics Inc. (Danvers, MA, USA) and operated at 2.6 MeV and 30 mA was commissioned in 2019 and should have met the conditions required for clinical trials by the present time [3,40].

In Japan in 2020, accelerator-based BNCT using the drug Borofalan^®^ containing boron phenylalanine (BPA) was approved for head and neck cancer treatment in adults, with national insurance coverage [4,5,44]. This has led to even greater public interest in the BNCT technology and a more thorough search for agents that can surpass BPA in efficacy [38,44,45,46].

In many countries, preclinical radiobiological studies are required prior to treating human malignancies with accelerator-based NCT. In this case, studies using cells and animal tumor models play a key role [45,47]. However, cell and tumor models in laboratory animals cannot fully repeat the complex morphology of spontaneous tumors. In addition, body weight, metabolic rate, and radiation tolerance in animals are different from those in humans.

In this case, enrollment of large animals of the mammal family, particularly domestic animals, such as cats and dogs, is the most suitable for estimating the radiobiological effects of NCT. Such animals have a large body mass, which allows for the selection of dosimetry parameters similar to those in humans. In addition, malignant tumors in large animals are easy to examine morphologically. The localization and nature of spontaneous tumors in domestic animals are similar to human tumors in many ways. Accordingly, the applied surgical treatment, chemotherapy, and radiation therapy methods show similar results. 

Since malignant tumors are a common cause of death in pets [48] and standard therapies do not always show satisfactory results, interest in using alternative therapies has increased. It has been shown previously that treating animals with spontaneous tumors using BNCT can be quite successful and justified [36,49]. Thus, conducting preclinical studies in large animals with spontaneous tumors is an important step to confirm the effectiveness of the method and potentially introduce it as a new treatment option in clinical practice.

NCT is designed to be applied mainly in head and neck cancer, intracerebral tumors, melanoma, and metastatic lesions, which are very common in pets. We are the first to conduct a preclinical GdNCT effectiveness study in large mammals with malignant tumors using a registered gadolinium-containing MRI contrast agent, Magnevist^®^ (dimeglumine gadopentetate), and a neutron beam obtained at the accelerator developed at the Institute of Nuclear Physics in Novosibirsk, Russian Federation [49] with irradiation conditions similar to those in our previous experiments on BNCT [50].

Thus, in this study, we aim to investigate the effect of GdNCT to increase the knowledge of NCT and optimize its therapeutic effectiveness. 

## 2. Materials and Methods

This study on gadolinium neutron capture therapy was performed based on ethical standards and approved by the Ethics Committee of the Institute of Cytology and Genetics, Novosibirsk, Russian Federation. In addition, the informed consent of the animal owners for this type of therapy was obtained in each case. For this study, we selected animals with spontaneous tumors in satisfactory somatic condition with the possibility of anesthesia and clinical observation for more than three months or those with tumors that were difficult or impossible to remove surgically, taking into account complications. We also included animals with tumors that had relapsed after surgical treatment or previous chemotherapy. A total of seven animals (five cats and two dogs) received GdNCT—the data are presented in Table 1.

Before treatment, a computed tomography (CT) scan was performed to visualize tumor location, size, and possible metastases. The imaging data were used for treatment planning. Before the GdNCT session, the neutron spectrum and the absorbed dose depth distribution in the simulated tumor, skin, and surrounding tissues were calculated. To calculate the minimum effective dose with a median of 30 Gy-Eq in the tumor, a tumor/surrounding tissues ratio of 3:1 was used, with an assumed average gadolinium concentration of 5.5 mg/g in the tumor at the time of irradiation and 1.8 mg/g in the surrounding tissues. The tumor-to-blood ratio was assumed to be 1:1. The upper dose limit for the skin was 18 Gy-Eq and 12 Gy-Eq for the mucosa [8]. Tumors were irradiated from both sides sequentially, with a change of position in the middle of the irradiation session, to ensure uniform exposure to the neutron flux.

The irradiation settings included a proton energy level of 2.05 MeV and an integral current between 2.2 and 4.4 mAh, depending on the presence of previous radiation therapy, tumor volume, and localization. The average irradiation session lasted for 2 h.

Dimeglumine gadopentetate (Magnevist^®^), manufactured by Bayer AG (Leverkusen, Germany), was administered fractionally at the maximum dose for humans recommended by the manufacturer (0.6 mL/kg). This was accomplished via an intravenous injection immediately before the radiation (0.3 mL/kg) and another injection (0.3 mL/kg) in the middle (at the half-time point) of the radiation session. Animals received intravenous anesthesia using a standard dose of Dexdomitor, dexmedetomidine plus Zoletil 100, or tiletamine plus zolazepam. 

After irradiation, 1 mL of peripheral vein blood was collected in a test tube for subsequent determination of the gadolinium concentration. Blood samples were analyzed using inductively coupled plasma atomic emission spectrometry (ICP-OES). The average blood concentration of gadolinium was 182 ± 217 ppm. Since this study was also performed to infer how much gadolinium accumulates within tumors, the amount of gadolinium in the tumors was predicted using an assumed tumor-to-blood ratio of 1:1 and a tumor-to-surrounding tissues ratio of 3:1.

Animal positioning during therapy, collection of blood samples for analysis, and post-treatment monitoring of the animals for three months were performed according to the methodology of the previous study [50].

## 3. Results

Animals (cats and dogs) are described in the sequence of their examination and treatment with GdNCT.

### 3.1. Female Cat “Ayuta”

Age: 11 years.Symptoms: worsening of nasal breathing and nasal discharge.Tumor localization: right nasopharyngeal area. The nasopharyngeal tumor was visualized using CT.Treatment other than GdNCT (surgery, radiation, or chemotherapy): none.Final pathological diagnosis: highly differentiated adenocarcinoma of the nasal mucosa.The course of the disease after GdNCT: during follow-up, the overall condition was satisfactory, and nasal breathing improved. At the examination, the visible part of the tumor decreased in size. A CT scan performed 3 months after GdNCT revealed no significant changes compared to the pretreatment visualization (Figure 1).

### 3.2. Male Dog “John”

Age: 14 years.Symptoms: impaired nasal breathing and nasal discharge.Localization and size of the tumor: nasal area; according to CT, the tumor size was 32 × 39 × 47 mm.Treatment other than GdNCT: surgical removal of the nasal tumor, twice.Final pathological diagnosis: chondrosarcoma.The course of the disease after GdNCT: after one month of follow-up observation, the overall condition was satisfactory. However, starting from the second month, a gradual deterioration of the condition, nasal bleeding, difficulty in breathing, and decreased appetite were noted (Figure 2).

### 3.3. Male Cat “Sausage”

Age: 6 years.Symptoms: impaired nasal breathing and nasal discharge.Tumor localization and size: left nasal area with bone destruction and regional lymphadenopathy. According to the CT scan, the tumor size was 33 × 26 × 26 mm.Treatment other than GdNCT: standard X-ray radiotherapy with doses of 35.4 Gy for the tumor (6 sessions) and 30 Gy for the lymph nodes (6 sessions). The outcome of that treatment was tumor recurrence.Pathological diagnosis: squamous cell carcinoma.The course of the disease after GdNCT: after irradiation, the animal was lethargic and had facial edema and a necrotic area (circled in the photo) with exudate. Two months after GdNCT, there were signs of tumor recurrence in the back of the nose, upper jaw, and corner of the eye (Figure 3).

### 3.4. Male Cat “Lev”

Age: 11 years.Symptoms: gingival swelling and ulceration, bleeding, decreased appetite, regional lymphadenopathy, and lacrimation from the right eye.Tumor localization and size: oral cavity, upper jaw on the right side, with metastasis to the submandibular lymph node. Based on a head CT, the tumor size was 25 × 40 × 15 mm.Treatment other than GdNCT: none.Pathological diagnosis: highly differentiated squamous cell carcinoma.The course of the disease after GdNCT: the animal’s condition and appetite after GdNCT were normal, and there was a visual reduction in tumor size during the early observation period. There was no discharge. After irradiation, the eye continued to tear. Twenty-three days after GdNCT, the condition worsened. The tumor began to disintegrate, the gum was covered with bloody ulcers, and holes appeared in the cavity; the cat also experienced spittle vomiting and black liquid stools, a decrease in appetite, and would rub the tumor site. After 42 days of treatment, the tumor had not become visually smaller, the gum was bleeding, the animal’s appetite was not quite stable, and constant use of analgesics was necessary. Three months after GdNCT, active tumor progression was found at the examination (Figure 4).

### 3.5. Female Dog “Lily”

Age: 13 years.Tumor localization and size: basal neoplasm of the middle lobe of the right lung with involvement of the cranial and caudal lobes and single metastases to the lungs and regional lymph nodes. According to the CT scan, the tumor size was 33 × 35 × 42 mm.Treatment other than GdNCT: none.Pathological examination was not done.The course of the disease after GdNCT: general improvement, breathing recovery, and right lung excursions one month after treatment. Activity and appetite improved. According to the CT scan, 2 months after GDNCT, the mass decreased in size to 32 × 38 × 36 mm (Figure 5).

### 3.6. Female Cat “Marusya”

Age: 10 years.Symptoms, localization, and size of the tumor: soft tissue mass in the upper jaw on the left side with invasion into the nasal cavity and left orbital space. Destruction of the upper jaw bone. According to the CT scan, the tumor size was 16 × 31 mm.Treatment other than GdNCT: none.Pathological diagnosis: squamous cell carcinoma.The course of the disease after GdNCT: the cat’s appetite improved, their eye was open a little, and their activity was unchanged. Sneezing disappeared after irradiation but then resumed at a lower frequency than before irradiation; grunting sounds (which were present before irradiation) also disappeared. The overall condition improved. One month after GdNCT, the condition worsened, with the tumor size increasing to 38 × 59 mm, according to CT. The condition further deteriorated, and subsequently, the animal died (Figure 6).

### 3.7. Male Cat “Semyon”

Age: 6 years.Localization and size of the tumor: right femur and iliac bone; according to a CT scan, the tumor was 34 × 32 × 37 mm in size.Treatment except for GdNCT: four courses of chemotherapy.Pathological diagnosis: osteogenic sarcoma.The course of the disease after GdNCT: on the tenth day after NCT, the animal was eating, drinking, and feeling well. A CT scan three months after irradiation revealed an increase in the tumor size to 58 × 38 × 46 mm (Figure 7).

## 4. Discussion

Previously, our research group conducted studies on BNCT in laboratory animal tumor models and spontaneous tumors in pets (cats and dogs) using BSH [45,50,51].

BSH and BPA are standard compounds typically used in BNCT. They are expensive and difficult to obtain in some countries. For example, a single clinical NCT session might require a volume of the drug that costs about 30,000 USD in equivalent. Since gadolinium compounds are currently used in clinical practice for contrast enhancement during MRI, such agents represent perfect theranostic drugs capable of combining NCT and MRI. Although this is an advantage of GdNCT, the use of ^157^Gd in NCT has been limited to cell experiments and small animal studies so far. Despite the advantages of gadolinium, such as a high gradient of accumulation ratio in tumors relative to healthy tissues, the large cross-section for thermal neutron absorption, and the possibility of using highly efficient Auger electrons from gadolinium to destroy tumor cells, there are several drawbacks associated with its use. During neutron capture by gadolinium, the most important issue is the effect of the generated gamma radiation on the healthy tissues surrounding the tumor. This radiation possesses high energy (the total energy of 7.9 MeV) and travels a considerable distance through biological tissues, which might cause irradiation of healthy cells and an insufficient irradiation dose to the tumor. Electrons and X-rays account for a small fraction of the energy released in a Gd(n, y) reaction, which is 63 keV. However, these electrons and X-rays can play a significant role in affecting the tumor, with their specific effects depending on the tumor’s size and the gadolinium location in relation to the tumor cell nuclei. Initially, many studies considered only the dose distribution from gamma radiation from Gd(n, y) reactions. However, Martin et al. (1989) and Laster et al. (1996) found that in their experiments with cells, the effect of gadolinium-related neutron capture reactions was more significant than identified in previous studies where it was associated only with gamma radiation [52,53]. In these studies, it was concluded that the additional effect was caused by the internal conversion of electrons and Auger electrons. However, for an exact explanation of this phenomenon, data on the dose distribution from all types of radiation arising in the reaction with gadolinium were needed. In subsequent studies, Cerullo et al. (2009) showed that according to dosimetry and microdosimetry of absorbed doses in both tumor and healthy tissues, gadolinium NCT appears to be a feasible treatment for malignant tumors [54].

Karpovich et al. proposed a method of therapy planning for GdNCT that relies on quantitative assessment of the content of Gd (III) complexes in the tumor and surrounding tissues based on a multicompartment phenomenological pharmacokinetic model adapted to real patient study data, with the possibility of visual assessment by MRI [55]. In patients receiving Magnevist^®^, there is a retention of gadolinium in the glial tumor, which is already clearly visible during the visual analysis of MR images, and at the same time, there are no changes in the intensity of the surrounding white matter. The ratios between the transport coefficients of Magnevist^®^ in the directions from blood to tissue and from tissue to blood were calculated. For glial tumor tissue, this ratio was 9.5–10, i.e., the reverse transfer was slower by one order of magnitude. Obviously, this situation will ensure sufficient retention time of Magnevist^®^ molecules in the tumor tissue with negligible accumulation in normal tissues, which is the key to successful NCT with selective tumor cell targeting [55]. These results were in good agreement with previously published quantitative autoradiography data [56].

Despite promising dosimetry data, the efficacy of Gd-DTPA and Gd-DOTA as therapeutic agents for GdNCT was initially predicted to be low due to a potentially insufficient accumulation of gadolinium in tumor cell nuclei. However, De Stasio et al. showed that Gd-DTPA penetrates cancer cell nuclei in vitro, which led to a reconsideration of the GdNCT approach using that compound [23]. In vivo studies have also suggested that multiple injections could lead to the penetration of Gd into more tumor cell nuclei [24]. 

To increase the effectiveness of Gd-DTPA in GdNCT, continuous intra-arterial infusions have been proposed [22]. In a study by Akine et al. (1993), Gd-DTPA was injected into the femoral artery on the side of a VX-2 tumor growing in the hind limbs of New Zealand white rabbits, which created a concentration of gadolinium 5–6 times higher than in a contralateral tumor on the opposite limb. At the site of injection, tumor growth was significantly suppressed between days 16 and 23 after Gd-NCT compared to control tumors [22]. The therapeutic effect of Gd-DTPA in vivo was also reported in rats with Jensen’s sarcoma [57]. 

A proof of principle for cell death by Gd neutron capture in U87 glioblastoma multiforme (GBM) cells preloaded with Magnevist^®^ has been presented [26]. Later, the authors described autophagy progression after neutron capture by gadolinium in U87 GBM cells [28].

Gadolinium NCT agents that are potentially more effective than Magnevist^®^ have also been studied. Brugger et al. (1989) published one of the first articles reporting on Gd chelates used as NCT agents [6]. The possibility of using Gd-DTPA and Gd-DOTA to treat brain tumors was investigated. It was demonstrated that these Gd derivatives could reach therapeutic concentrations in brain tumors. In addition, a good ratio of Gd concentration in tumor tissue compared to normal tissues was observed [6]. Matsumura et al. (2003) demonstrated the efficacy of the drug Gd-BOPTA (Multihance^®^) as an agent for neutron capture therapy [58]. Gd-BOPTA showed twice as long accumulation in tumors compared to other drugs used for MR imaging.

Different researchers suggested using liposomal compositions or chitosan to increase the concentration and long-term deposition of gadolinium in tumors [25,27,29]. Le et al. studied the biodistribution and accumulation of Gd-DTPA-encapsulated liposomes in tumor-bearing mice [25]. The amount of Gd accumulated in the tumors in this study not only exceeded the calculated Gd concentration required for successful NCT but also reached an average value of 158.9 ± 48.7 µg/g 12 h after intravenous injection [25]. 

Fujimoto T. et al. (2009) used gadopentecic acid (Gd-DTPA) with chitosan to develop gadolinium-loaded chitosan nanoparticles (Gd-nanoCPs) and studied their accumulation and MR imaging visualization enhancement effect compared to Gd-DTPA (Magnevist^®^) in human malignant fibrosis histiocytoma (MFH) cells [27]. The authors showed more efficient Gd accumulation after using Gd-nanoCPs; however, the longitudinal relaxation time (T1) reduction was more prominent in the case of Gd-DTPA [27]. In a later study by the same research group, Ichikawa et al. (2014) used Gd-nanoCPs for controlled Gd delivery in NCT [29]. In vitro experiments were performed using a melanoma cell line; mice with a subcutaneous tumor xenograft were also used. In vitro, smaller Gd nanoparticles accumulated in cells in higher concentrations and were thus more suitable for NCT. Nanoparticles incorporating 1.2 mg of natural Gd were injected once or twice intratumorally into subcutaneous B16F10 melanoma-bearing mice. Eight hours after the last injection, the tumor areas of the mice were irradiated with thermal neutrons. The Gd-NCT group showed significant tumor growth suppression, although nanoparticle effectiveness was found to depend on their micrometric properties [29]. Other authors have proposed using hybrid complexes with MR imaging capabilities, including gadolinium chelates, to treat tumors with NCT [59].

Regarding studies on neutron capture therapy in large animals using gadolinium compounds, a group led by Professor Mitin treated dogs aged between 9 and 15 years with histologically confirmed diagnoses of oral mucosal melanoma [60]. Dipentast^®^ and BPA were used. In the GdNCT group, 70% of cases achieved complete regression of the tumor within 30–45 days; in 14% of the animals, recurrence occurred after one month, and life expectancy was between four and six months. In the BNCT group, complete tumor regression was achieved in 95% of cases; in 20% of cases, recurrence occurred after three to four months, and the life expectancy was approximately eight months. In the gamma therapy group, 90% of the dogs showed tumor recurrence one month after treatment, metastases after 2–3 months, and a life expectancy of approximately three months. 

In our study, according to the Magnevist^®^ drug instructions, we used the maximum recommended dosage of 0.6 mL per kg of body weight or 0.3 mmol per kg of body weight, which in a proportional ratio should provide up to 180 µg of ^157^Gd/g in tumor tissue. We injected the drug twice at half the dose calculated per body weight to ensure a homogeneous concentration of the drug in the tumor for the entire duration of the GdNCT session.

Despite meeting all the conditions for maximum accumulation of the drug in the tumor tissue, which suggests a maximum NCT effect, we observed negative changes that manifested as increases in tumor volume and clinical deterioration during the follow-up period one to three months after GdNCT in most of the animals. We observed a short-term absence of disease progression only in some cases; there were also no adverse reactions of the skin or fur at the irradiated sites. 

Thus, we could not show a significant positive effect of NCT for spontaneous malignant tumors in domestic animals. Computed tomography performed three months after GdNCT showed no changes or increases in tumor volumes, indicating an insufficient tumor response in all cases. Survival with poor quality of life was observed up to three months after irradiation, and the survival rate and quality of life did not improve over time. Typically, within two weeks (14 days) after GdNCT, the initial positive treatment response was replaced by a deterioration of the overall condition and quality of life. No change in tumor size after therapy and continued tumor growth could be related to the spread, depth, and infiltrative nature of spontaneous tumors as well as an insufficient concentration and rapid washout of Gd-DTPA from tumor tissues despite two injections during irradiation sessions, the extracellular localization of Gd-DTPA, and the presence of Gd-DTPA mainly in the tumor periphery. As a result, large tumors received insufficient radiation doses in the central and deeper parts. Thus, the small number of animals, different morphological characteristics of the tumors, and the lack of computed tomography data in some cases may represent the current study’s limitations.

A solution to improve the therapeutic outcome at this stage would be using gadolinium drugs that can accumulate inside tumor cells, performing several GdNCT sessions, and combining GdNCT with other treatment modalities. Therefore, it is obvious that new gadolinium-containing agents capable of selective accumulation in tumors are required for GdNCT. Based on recent publications, one of the most promising theranostic agents is AGuIX, a radiosensitizer and MRI contrast agent with a high gadolinium concentration in its structure [61]. It has been employed in phase II clinical trials in patients with brain metastases and in phase I trials in patients with glioblastoma, lung, cervical, and pancreatic cancer [61]. Other newly synthesized gadolinium nanoparticles have recently been introduced, and further development of GdNCT is in progress [62,63].

## 5. Conclusions

Neutron capture therapy often remains the last option for the treatment of spontaneous tumors in animals, while other methods prove to be ineffective. Studies on large animals allow for treatment conditions to be adjusted as close as possible to those of clinical trials and help clarify the technical aspects of NCT. Using a unique accelerator source for neutron generation, as done in these studies, represents a pioneering application of the method worldwide. The combination of gadolinium administration and accelerator-based neutron irradiation provides additional data on the current conditions of GdNCT. These data can then be compared with those obtained by scientific groups conducting BNCT and other methods of radiation therapy for malignant tumors. At this stage, it is evident that successful GdNCT requires new gadolinium-containing drugs capable of sufficient tumor accumulation. Furthermore, the results of preclinical studies in large animals will allow us to prepare for the initiation of appropriate clinical trials.

## Figures and Tables

**Figure 1 vetsci-10-00274-f001:**
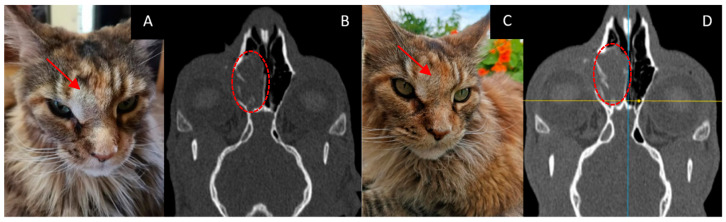
Female cat “Ayuta” (11 years old) with highly differentiated adenocarcinoma in the right nasopharyngeal area, leading to worsening of nasal breathing and nasal discharge. Tumor appearance ((**A**), arrow) and head CT ((**B**), dashed circle) before irradiation, and tumor appearance ((**C**), arrow) and head CT ((**D**), dashed circle) 3 months after GdNCT. No significant changes compared to the pretreatment visualization were revealed.

**Figure 2 vetsci-10-00274-f002:**
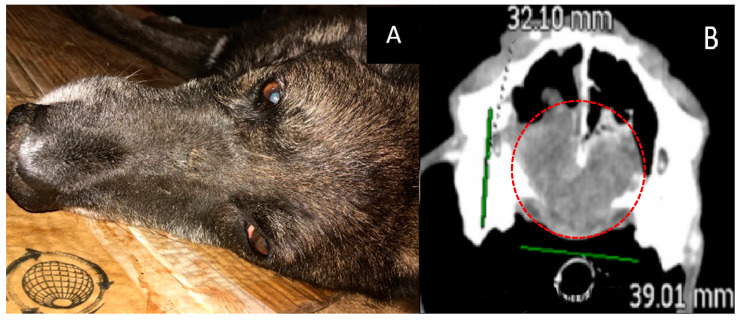
Male dog “John” (14 years old) with chondrosarcoma in the nasal area with impaired nasal breathing and nasal discharge and tumor recurrence after repeated surgical treatment. Animal appearance before irradiation (**A**). Head CT after irradiation (**B**) with a tumor 32 × 39 × 47 mm in size (green lines, dashed red circle).

**Figure 3 vetsci-10-00274-f003:**
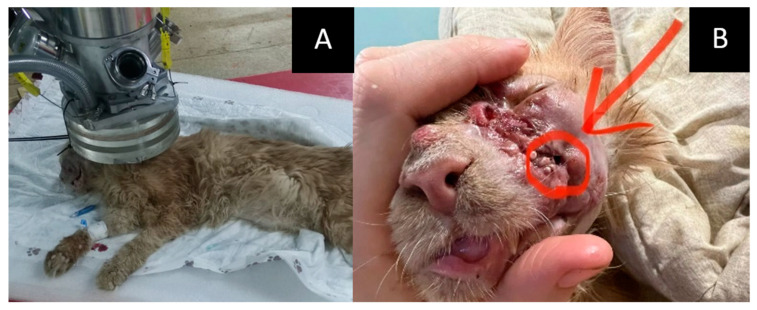
Male cat “Sausage” (6 years old) with squamous cell carcinoma in the left nasal area (33 × 26 × 26 mm) with bone destruction, causing impaired nasal breathing and nasal discharge, and regional lymphadenopathy. Animal positioning under the lithium target within the beam shaping assembly during neutron irradiation (**A**) and tumor appearance after the irradiation ((**B**), arrow and the circle). Two months after GdNCT, there were signs of tumor recurrence in the back of the nose, upper jaw, and corner of the eye.

**Figure 4 vetsci-10-00274-f004:**
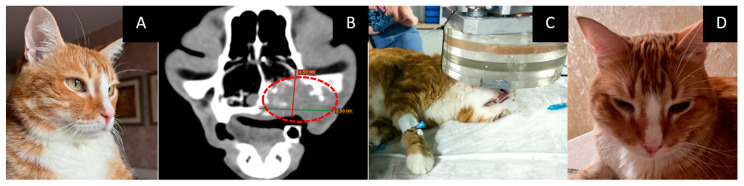
Male cat “Lev” (11 years old) with highly differentiated squamous cell carcinoma of the oral cavity and upper jaw on the right side and metastasis to the submandibular lymph node. Animal appearance (**A**) and head CT (**B**) before irradiation with a tumor in the oral cavity 25 × 40 × 15 mm in size (green and red lines, dashed red circle). Animal positioning under the lithium target during GdNCT (**C**) and animal appearance after irradiation (**D**). Active tumor progression was found three months after irradiation.

**Figure 5 vetsci-10-00274-f005:**
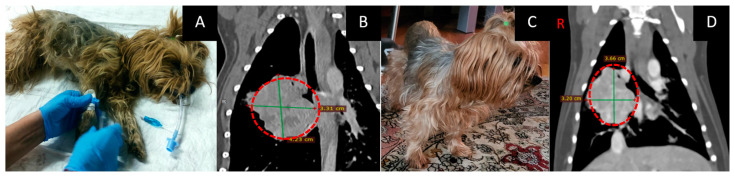
Female dog “Lily” (13 years old) with a tumor of unknown origin (pathological examination was not done) in the middle lobe of the right lung with involvement of the cranial and caudal lobes and single metastases to the lungs and regional lymph nodes Animal preparation for GdNCT (**A**). Chest CT (**B**) before irradiation showed a tumor 33 × 35 × 42 mm in size (green lines, dashed red circles). Animal appearance after irradiation (**C**). Two months after GDNCT, chest CT scans (**D**) showed that the tumor mass decreased in size to 32 × 38 × 36 mm (dashed line).

**Figure 6 vetsci-10-00274-f006:**
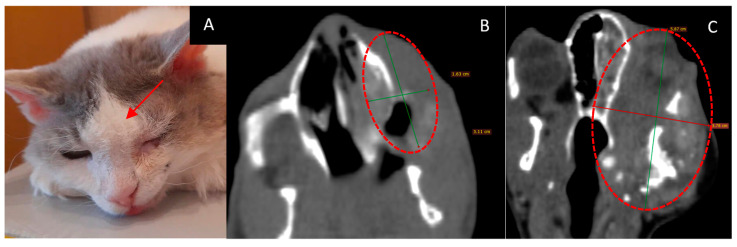
Female cat “Marusya” (10 years old) with squamous cell carcinoma in the upper jaw on the left side with invasion into the nasal cavity and left orbital space and destruction of the upper jawbone. Animal appearance before irradiation (**A**) with the outer tumor mass (arrow). Head CT (**B**) before irradiation showed a tumor 16 × 31 mm in size (green lines, dashed red circle). Head CT after the treatment (**C**) showed that the tumor had increased in size to 38 × 59 mm (green and red lines, dashed red circle).

**Figure 7 vetsci-10-00274-f007:**
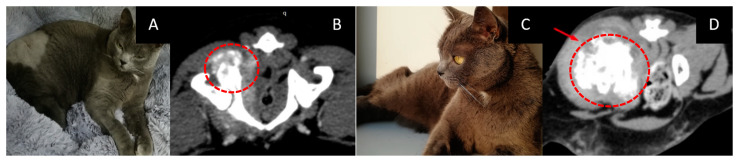
Male cat “Semyon” (6 years old) with osteogenic sarcoma of the right femur and iliac bone. Animal appearance before irradiation (**A**). CT scan before irradiation (**B**) showed a tumor 34 × 32 × 37 mm in size (dashed circle). Animal appearance after irradiation (**C**). CT scan three months after irradiation (**D**) revealed that the tumor had increased in size to 58 × 38 × 46 mm (arrow, dashed circle).

**Table 1 vetsci-10-00274-t001:** Animal data, including each animal’s type, sex, age, tumor location and pathology, and irradiation parameters.

№	Type	Age, Years	Name	Localization of the Tumor	Pathological Diagnosis	Integral Current
1	cat	11	Ayuta	nasopharynx	highly differentiated adenocarcinoma	4
2	dog	14	John	nose	chondrosarcoma	4.4
3	cat	6	Sausage	nose with bone destruction,	squamous cell carcinoma	2.2
4	cat	11	Leva	gum and jaw lymphadenopathy	highly differentiated squamous cell carcinoma	4.4
5	dog	13	Lily	lung cancer, lung metastases	–	3.4
6	cat	10	Marusya	oral cavity, jaw	squamous cell carcinoma	4
7	cat	7	Semyon	femur and ilium	sarcoma	4

## Data Availability

The data presented in this study are available upon request from the corresponding author.

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
