# Peer review of "Gadolinium Neutron Capture Therapy for Cats and Dogs with Spontaneous Tumors Using Gd-DTPA"

_vetsci, 2023, doi:10.3390/vetsci10040274_

Round 1

Reviewer 1 Report

The authors propose gadolinium NCT (GdNCT) for veterinary cancer therapy in particular for end stage head and neck cancers in pets. The manuscript is well presented and clearly written, the case studies well documented. This referee suggests a few lines in the introduction that define the state of the art in human medicine that highlights the problems that have emerged and the scientific discussion in this regard. 

Author Response

We thank the Reviewer for a positive comment and essential suggestion that helped us considerably improve the manuscript.

We have added the following information regarding problems of invasive cancers that can be solved by neutron capture therapy (page 2, lines 46-54), added more details about clinical trials on boron neutron capture therapy (page 2, lines 76-81), and modified the paragraph on boron compounds limitations (page 2, lines 82-86):

"Certain malignancies are extremely difficult to treat and still remain lethal due to their invasive nature and the inability of all modern treatment modalities to influence single tumor cells that migrate into healthy tissues and lead to the development of new tumor masses at a distance from the initial tumor that seemed to be cured. This may happen in the case of malignant melanoma with distant metastases, malignant glioma with fast local recurrence, or another type of invasive cancer that leaves no chance of survival despite the best efforts of modern medicine. Therefore, there is still a search for more effective cancer therapies, including more advanced irradiation methods capable of targeting single tumor cells."

" Thus, Fukuda et al. have shown the effectiveness and limitations of BNCT for malignant melanoma [8], Kato et al. [9] and Kankaanranta et al. [12] reported on initial clinical trials on BNCT for recurrent head and neck cancer, Japanese research groups, including Yamamoto et al. [11] and Miyatake et al. [13], demonstrated the efficacy of the method in patients with malignant brain tumors, and Wittig et al. showed the ability of 10B compounds to accumulate in liver metastases of colorectal cancer [10]."

" However, the efficacy limitations of previously and currently used boron compounds, the complexity of the boron enrichment process for their production, and their high cost restrict the use of 10B-containing compounds in research and clinical practice. Therefore, Gd-based drugs, already registered in many countries and used clinically for purposes other than NCT, could be an alternative to 10B compounds."

The state-of-art accelerator NCT projects have been additionally described as follows (page 3, lines 105-119):

“Among the most advanced projects, we can mention the BNCT System NeuCure® by Sumitomo Heavy Industries, Ltd. (Tokyo, Japan), based on a cyclotron accelerator operated at the energy of 30 MeV and the current of 2 mA and a beryllium target, which is already installed in clinical treatment facilities in Japan [3, 5, 32]. A tandem accelerator with vacuum insulation and a solid lithium target has been developed at the Budker Institute of Nuclear Physics in Novosibirsk, Russian Federation, and later, together with TAE Life Sciences (Foothill Ranch, CA, USA), modified and installed in Xiamen Humanity Hospital (Xiamen, Fujian, China) [3]. The University of Tsukuba, together with Mitsubishi Heavy Industry Co., Ltd. and several other organizations, have developed a linear accelerator with a beryllium target that has been tested preclinically with planned initiation of clinical trials in 2023 [3, 33]. A BNCT center in Helsinki, Finland, with electrostatic accelerator Hyperion™ with a rotating lithium target developed by Neutron Therapeutics Inc. (Danvers, MA, USA) and operated at 2.6 MeV and 30 mA was commissioned in 2019 and should have the conditions for clinical trials by the present time [3, 40].”

Reviewer 2 Report

1) Brief Summary: Gadolinium neutron capture therapy for cats and dogs with spontaneous tumors using Gd-DTPA

            The authors intended "to compare the effect of GdNCT (gadolinium neutron capture therapy) with that of BNCT (boron neutron capture therapy) in a similar animal group in our previous study to expand the knowledge of NCT further and optimize its therapeutic effectiveness."

            To achieve their goals, the authors used five cats and two dogs, with various malignant tumors. The animals were treated with dimeglumine gadopentetate (Magnevist®, Bayer AG, Germany, administered fractionally in the maximum dose for humans recommended by the manufacturer (0.6 ml/kg): immediately before the radiation, intravenously (0.3 ml/kg) and repeated (0.3 ml/kg) in the middle (at half-time) of the radiation session.

            After irradiation, peripheral vein blood was taken in a volume of 1 ml into a test tube for subsequent determination of gadolinium concentration by inductively coupled plasma atomic emission spectrometry.

2) General concept comments

            This investigation is very interesting and informative. However, the objective set forth by the authors to compare this work with previous published data "in a similar animal group" is not only unnecessary, but also inappropriate and inaccurate.

I. The Title. The title is adequate, considering the work performed by the authors.

II. The Introduction itself is appropriate and informative. Nevertheless, the objective of comparing this gadolinium capture therapy study with a previous one that applied boron neutron capture therapy, in my opinion, must be stripped from the manuscript. It is desirable that the authors use their previous, analogous work, to discuss the results of the present investigation; however, their claim that this study was a comparison between similar groups of animals, is not accurate. The tumors on the animals were different, the results were presented in detail only for the GdNCT technique, the discussion was (correctly) centered on the GdNCT results. Thus, the appropriate aim would be, "we aim to investigate the effect of GdNCT to expand the knowledge of NCT further and optimize its therapeutic effectiveness."

III. The Materials and Methods

            The Materials and Methods are concise and adequate.

IV. The Results

a) All figure legends are not self-explanatory. It is very difficult to identify the features displayed (for example, "tumor appearance") on the images. Please expand all figure legends to make the figures entirely self-explanatory for the reader.

b) Specific features must be indicated in all figures with arrows, asterisks, and other markings. Interestingly, Figure 3B already follows that procedure, even though the arrow could be less conspicuous.

V. The Discussion

            The discussion is adequate. The authors emphasize their main findings, highlighting the importance and the limitations of the study, along with the implications and future paths to follow.

VI. The Conclusions

            I recommend that the second paragraph must be stricken out of the text, since it is related to the objective of comparison between GdNCT and BNCT data. The first and third paragraphs are adequate.

References

            The references are updated; 24 out of 64 references have been published in the last five years.

Author Response

1) Brief Summary: Gadolinium neutron capture therapy for cats and dogs with spontaneous tumors using Gd-DTPA

            The authors intended "to compare the effect of GdNCT (gadolinium neutron capture therapy) with that of BNCT (boron neutron capture therapy) in a similar animal group in our previous study to expand the knowledge of NCT further and optimize its therapeutic effectiveness."

            To achieve their goals, the authors used five cats and two dogs with various malignant tumors. The animals were treated with dimeglumine gadopentetate (Magnevist®, Bayer AG, Germany, administered fractionally in the maximum dose for humans recommended by the manufacturer (0.6 ml/kg): immediately before the radiation, intravenously (0.3 ml/kg) and repeated (0.3 ml/kg) in the middle (at half-time) of the radiation session.

            After irradiation, peripheral vein blood was taken in a volume of 1 ml into a test tube for subsequent determination of gadolinium concentration by inductively coupled plasma atomic emission spectrometry.

Response: We thank the Reviewer for such a thorough analysis of our manuscript and for critical comments and helpful suggestions. We have taken these comments into account, and they helped to improve our manuscript considerably.

2) General concept comments

            This investigation is very interesting and informative. However, the objective set forth by the authors to compare this work with previous published data "in a similar animal group" is not only unnecessary, but also inappropriate and inaccurate.

Response: We thank the Reviewer for this critical comment. We have modified the text accordingly.

  1. The Title. The title is adequate, considering the work performed by the authors.

Response: We thank the Reviewer for the positive comment.

  1. The Introduction itself is appropriate and informative. Nevertheless, the objective of comparing this gadolinium capture therapy study with a previous one that applied boron neutron capture therapy, in my opinion, must be stripped from the manuscript. It is desirable that the authors use their previous, analogous work, to discuss the results of the present investigation; however, their claim that this study was a comparison between similar groups of animals, is not accurate. The tumors on the animals were different, the results were presented in detail only for the GdNCT technique, the discussion was (correctly) centered on the GdNCT results. Thus, the appropriate aim would be, "we aim to investigate the effect of GdNCT to expand the knowledge of NCT further and optimize its therapeutic effectiveness."

Response: We thank the Reviewer for this critical comment and suggestion. We have modified the Introduction accordingly.

“NCT is designed to be applied mainly in head and neck cancer, intracerebral tumors, melanoma, and metastatic lesions, which are very common in pets. We are the first to conduct a preclinical GdNCT effectiveness study in large mammals with malignant tumors using a registered gadolinium-containing MRI contrast agent, Magnevist® (dimeglumine gadopentetate), and a neutron beam obtained at the accelerator developed at the Institute of Nuclear Physics in Novosibirsk, Russian Federation [49] with irradiation conditions similar to those in our previous experiments on BNCT [50].

Thus, in this study, we aim to investigate the effect of GdNCT to further expand the knowledge of NCT and optimize its therapeutic effectiveness.”

III. The Materials and Methods

            The Materials and Methods are concise and adequate.

Response: We thank the Reviewer for the positive comment.

  1. The Results

  1. a) All figure legends are not self-explanatory. It is very difficult to identify the features displayed (for example, "tumor appearance") on the images. Please expand all figure legends to make the figures entirely self-explanatory for the reader.

  1. b) Specific features must be indicated in all figures with arrows, asterisks, and other markings. Interestingly, Figure 3B already follows that procedure, even though the arrow could be less conspicuous.

Response: We thank the Reviewer for this critical comment. We have added more information to figure legends according to this helpful suggestion.

  1. The Discussion

            The discussion is adequate. The authors emphasize their main findings, highlighting the importance and the limitations of the study, along with the implications and future paths to follow.

Response: We thank the Reviewer for the positive comment.

  1. The Conclusions

            I recommend that the second paragraph must be stricken out of the text, since it is related to the objective of comparison between GdNCT and BNCT data. The first and third paragraphs are adequate.

Response: We have rewritten the conclusion according to the Reviewer’s recommendations:

“Neutron-capture therapy often remains the last option for the treatment of spontaneous tumors in animals, while other methods prove to be ineffective. Studies on large animals allow for adjusting the treatment conditions as close as possible to those of clinical trials and help clarify technical aspects of NCT. Using a unique accelerator source for neutron generation makes such studies a pioneering application of the method worldwide. The combination of gadolinium administration and accelerator-based neutron irradiation provides additional data on the current conditions of GdNCT, comparing the results with the data of other scientific groups conducting BNCT as with other methods of radiation therapy for malignant tumors. At this stage, it is evident that successful GdNCT requires new gadolinium-containing drugs capable of sufficient tumor accumulation. And the results of preclinical studies in large animals will allow us to prepare for the initiation of appropriate clinical trials.”

References

            The references are updated; 24 out of 64 references have been published in the last five years.

Response: We thank the Reviewer for such a thorough literature analysis and are glad that our manuscript was based on the updated literature data.
